# [Re] GNNInterpreter: A probabilistic generative model-level explanation for Graph Neural Networks

**Ana-Maria Vasilcoiu**                                                      *ana-maria.vasilcoiu@student.uva.nl*
*University of Amsterdam*

**Thijs Stessen**                                                                   *thijs.stessen@student.uva.nl*
*University of Amsterdam*

**Thies Kersten**                                                                  *thies.kersten@student.uva.nl*
*University of Amsterdam*

**Batu Helvacioğlu**                                                          *batu.helvacioglu@student.uva.nl*
*University of Amsterdam*

**Reviewed on OpenReview:** *https://openreview.net/forum?id=8cYcR23WUo*

## Abstract

Graph Neural Networks have recently gained recognition for their performance on graph machine learning tasks. The increasing attention on these models' trustworthiness and decision-making mechanisms has instilled interest in the exploration of explainability techniques, including the model proposed in "GNNInterpreter: A probabilistic generative model-level explanation for Graph Neural Networks." (Wang & Shen (2022)). This work aims to reproduce the findings of the original paper, by investigation the main claims made by its authors, namely that GNNInterpreter (i) generates faithful and realistic explanations without requiring domain-specific knowledge, (ii) has the ability to work with various node and edge features, (iii) produces explanations that are representative for the target class and (iv) has a much lower training time compared to XGNN, the current state-of-the-art model-level GNN explanation technique. To reproduce the results, we make use of the open-source implementation and we test the interpreter on the same datasets and GNN models as in the original paper. We conduct an enhanced quantitative and qualitative evaluation, and additionally we extend the original experiments to include another real-world dataset with larger graphs. Our results show that the first claim can only be verified for the dataset with larger graphs. We reject the first claim for all other datasets, due to significant seed variation and training instability, as well as extensive need for hyperparameter tuning and domain knowledge. Furthermore, we partially validate the second claim by testing on datasets with different node and edge features, but we reject the third claim due to GNNInterpreter's failure to outperform XGNN in producing dataset aligned explanations. Lastly, we are able to confirm the last claim.

## 1 Introduction

Graph Neural Networks (GNNs) have recently surfaced as powerful tools for modeling graph-structured data, demonstrating cutting-edge performance in various applications, such as graph classification (Xu et al. (2018), Zhang et al. (2018)), node classification (Veličković et al. (2017)) and link prediction (Zhang & Chen (2018)). However, as these models gain prominence, the scrutiny on their trustworthiness and their decision-making mechanisms intensifies, especially in high-stakes domains such as chemistry or biomedicine where inaccurate predictions can have substantial consequences.

This has instilled a lot of interest in the exploration of explainability techniques for graph models. Compared to models that work on text or image data, there are considerably more challenging obstacles that need to be addressed (Yuan et al. (2022)): discrete adjacency matrices cannot be optimized via gradient-based methods (Duval & Malliaros (2021)), domain knowledge is often necessary and graph data is heterogeneous and highly complex. Despite these challenges, two types of graph explainability methods have emerged, each with several potent techniques: instance-level explanations (Luo et al. (2020a), Ying et al. (2019b), Vu & Thai (2020) and model-level explanations (Yuan et al. (2020)).

Model-level explanations have been shown to possess considerable advantages over their counterparts, with the XGNN (Yuan et al. (2020)) being the current state-of-the-art such model. A more recent work however, "GNNInterpreter: A probabilistic generative model-level explanation for Graph Neural Networks" (Wang & Shen (2022)) proposes a novel approach that claims to be more general, flexible and computationally efficient. In this work, we will analyze the findings presented in this paper and verify its claims, as well as present additional experiments that provide further evidence to our analysis. The code for running the reproduced experiments as well as our extensions can be found at https://anonymous.4open.science/r/Reproduction_GNNInterpreter/README.md.

## 2 Scope of reproducibility

This study aims to examine and validate the results demonstrated by Wang & Shen (2022) through reproduction of their experiments. Our main goal is to verify the claims of the original paper, which can be summarized as follows:

- Claim 1: The explanations generated by GNNInterpreter are faithful and realistic. Additionally, GNNInterpreter doesn't require domain-specific knowledge to achieve that.
- Claim 2: GNNInterpreter is a general approach that performs well with different types of node and edge features.
- Claim 3: The explanations generated by GNNInterpreter are more representative regarding the target class compared to XGNN.
- Claim 4: The time complexity for training GNNInterpreter is much lower than for XGNN.

Our contributions in regards to verifying these claims and additional findings are as follows:

- Our further analysis on the training dynamics of GNNInterpreter demonstrates that the method performs best on large graphs, but experiences training instability when working with small graphs.
- We investigate the impact of graph size on how faithful and realistic the explanations generated by GNNInterpreter tend to be. We find evidence that larger graphs tend to produce more faithful and realistic results.
- We find evidence that the GNNInterpreter requires hyperparameter tuning with domain-specific knowledge or searching random initializations to generate faithful and realistic explanations when used with datasets that have small graphs.
- We show evidence that GNNInterpreter can be used as a general approach with different types of node and edge features.
- Based on our qualitative analysis, we show that the explanations generated by GNNInterpreter are generally on par with XGNN and are outperformed by GNNExplainer for certain datasets.
- We show empirically that the time complexity of GNNInterpreter is much lower than XGNN, but that in practice the time spent for hyperparameter tuning or random seed searching should be accounted for.

## 3 Background

**Graph Neural Networks (GNNs).** GNNs have demonstrated remarkable progress in recent years, addressing challenges associated with diverse types of graph-based data. Despite the plethora of proposed models (Kipf & Welling (2017), Gilmer et al. (2017), Veličković et al. (2017)), they often converge on a shared concept of message passing. Due to their success and ubiquity in various applications, as well as due

to their increased complexity, the need for interpretability and explainability has become more pronounced than ever. As such, numerous post-hoc techniques that aim to explain the inner workings of GNN models have emerged, with two main categories, namely instance-level and model-level explanations.

**Instance-Level Explanation of GNNs.** Explaining GNNs at an instance level has recently gained momentum, with a goal of explaining particular predictions for specific inputs one at a time. With some exceptions, GNN instance explanation methods can broadly be classified into six categories, including gradient-based (Baldassarre & Azizpour (2019)), perturbation-based(Yuan et al. (2021), Ying et al. (2019a), Luo et al. (2020b)), decomposition(Schwarzenberg et al. (2019)), surrogate(Duval & Malliaros (2021)), generation-based(Shan et al. (2021), Lin et al. (2021)) and counterfactual-based methods. One noteworthy example in this category of models is the GNNExplainer (Ying et al. (2019a)) which works by identifying the most important subgraph structure and node features in that subgraph that contribute to the GNN's prediction.

**Model-Level Explanation of GNNs.** In contrast with local explanations, model-level explanations can give insight into how a black box machine learning model makes decisions as a whole, irrespective of any particular input. For tabular data, this can be achieved using techniques such as partial dependence plots (Friedman (2001), Hooker (2007),Apley & Zhu (2020)) or permutation feature importance (Strobl et al. (2008), Strobl et al. (2007), Apley & Zhu (2020)). However, these techniques are not applicable to GNNs as the intricate nature of graph configurations doesn't allow for straightforward modifications. For explaining graph classification, one approach that can deal with the variability of graph data is the generation of explanation graphs, namely prototypical examples for a given class. This allows for the opportunity of analyzing what graph patterns or sub-graphs patterns lead to certain predictions. In turn, this gives high-level insights about what the GNN model is focusing on and a better understanding of whether the model works as expected and otherwise how it can be adjusted.

**GNNInterpreter, XGNN and GCExplainer.** GNNInterpreter, the focus of the original paper that is being reproduced in this study and one approach to model-level explanation, achieves its purpose by training a generative model, while relaxing the discrete nature of edges. Further details about this can be found in Section 4.3. Another such model and the current state-of-art approach for graph explainability is XGNN Yuan et al. (2020), which in contrast formulates the graph generation problem as a reinforcement learning task. In this case, the generator predicts how to add an edge into the current graph and it is trained to generate graphs that maximize the class score of the target class by using a policy gradient method based on information from the trained GNN. Ensuring the validity and intelligibility of explanation graphs is done by incorporating graph rules, which may include simple constraints such as enforcing a single edge between any two nodes or more complex domain specific rules (e.g. chemical valency check in a molecular graphs dataset). Lastly, a noteworthy post-hoc model-level explanation method is GCExplainer (Magister et al. (2021)), which adapts the well-known Automated Concept-based Explanation approach (Ghorbani et al. (2019)) and puts the human in the loop to ensure the relevance and comprehensibility of its explanations.

**Self-interpretable GNNs.** In response to post-hoc methods failing to reveal the original reasoning of GNNs, the pursuit of self-interpretable GNN models, which integrate explainability into their architecture, has also recently gained traction (Kakkad et al. (2023)). Some examples include: (1) ProtGNN (Zhang et al. (2022)) that derives explanations from a case-based reasoning process and generates predictions by comparing inputs to learned prototypes in the latent space and (2) the model introduced by Dai & Wang (2021) which explains node classification tasks based on a K-nearest approach. Worth mentioning is that self-interpretability is also not ideal, and it often comes at the cost of model accuracy.

## 4 GNNInterpreter model

GNNInterpreter is a model-agnostic and model-level explanation method aiming to reveal the high-level decision-making process of message passing based GNNs. It works by learning a probabilistic generative graph distribution in order to produce the most discriminative graph pattern that the explained GNN detects when making a prediction.

**Learning objective.** GNNInterpreter achieves its goal by numerically optimizing a novel objective function with two distinct but important parts: (1) maximizing the likelihood of the explanation graph to be predicted

as the target class by the GNN model and (2) confining the explanation graphs distribution within domain-specific boundaries. Instead of manually defining domain rules, GNNInterpreter leverages the abstract knowledge learned by the GNN and achieves its second goal by maximizing the similarity between the embedding of the explanation graph and the average embedding of all target class graphs from the training data. Mathematically, this can be formulated as follows:

$$\max_{G} L(G) = \max_{A,Z,X} L(A,Z,X) = \max_{A,Z,X} \phi_c(A,Z,X) + \mu \cdot \mathrm{sim}_{\cos}(\psi(A,Z,X), \bar{\psi}_c) \tag{1}$$

where L is the objective function; A represents the adjacency matrix, Z the edge feature matrix and X the node feature matrix; $\phi_c$ is the scoring function corresponding to the target class c; $\psi$ is the graph embedding function of the explained GNN and $\bar{\psi}_c$ is the average graph embedding of c class graphs; $\mathrm{sim}_{\cos}$ denotes the cosine similarity; and lastly, $\mu$ is a hyper-parameter describing the weight factor.

**Graph distribution.** To learn a probabilistic graph distribution that will try to maximize the target class, two assumptions are being made. Firstly, the explanation graph is assumed to be a Gilbert random graph (Gilbert (1959)). Secondly, the features are independently distributed, allowing for the following factorization of a random graph variable $G$:

$$P(G) = \prod_{i \in \mathbf{V}} P(x_i) \cdot \prod_{(i,j) \in \mathbf{E}} P(z_{ij}) P(a_{ij}) \tag{2}$$

where $x_i$ represents the node feature for node $i$, $z_{ij}$ the edge feature between $i,j$ and $a_{ij} = 1$ if there is en edge $(i,j)$. The variables were modeled to be drawn from distributions. $a_{ij} \sim \mathrm{Bernoulli}(\theta_{ij})$, where $\theta_{ij}$ is the probability of edge $(i,j)$ existing. $z_{ij} \sim \mathrm{Categorical}(\mathbf{q}_{ij})$ and $x_i \sim \mathrm{Categorical}(\mathbf{p}_i)$ where $\|\mathbf{q}_{ij}\|_1 = \|\mathbf{p}_i\|_1 = 1$. The learning objective can thus be rewritten to maximize it over those random variables:

$$\max_{G} L(G) = \max_{\Theta,Q,P} \mathbb{E}_{G \sim P(G)} \left[ \phi_c(\mathbf{A}, \mathbf{Z}, \mathbf{X}) + \mu \cdot \mathrm{sim}_{\cos}(\psi(\mathbf{A}, \mathbf{Z}, \mathbf{X}), \overline{\psi}_c) \right] \tag{3}$$

**Continuous relaxation.** To address the discrete nature of graphs and achieve node and edge feature flexibility, GNNInterpreter employs continuous relaxation of the discrete variables to continuous variables which are then specified using the Concrete distribution (Maddison et al. (2016)). The random variables are modeled to be continuous as follows:

$$\begin{cases} \tilde{x}_i \sim \mathrm{Concrete}(\xi_{ij}, \tau_x) & \text{for } \tilde{x}_i \in \tilde{\mathbf{X}} \text{ and } \xi_i \in \mathbf{\Xi} \\ \tilde{z}_{ij} \sim \mathrm{Concrete}(\eta_{ij}, \tau_z) & \text{for } \tilde{z}_{ij} \in \tilde{\mathbf{Z}} \text{ and } \eta_{ij} \in \mathbf{H} \\ \tilde{a}_{ij} \sim \mathrm{BinaryConcrete}(\omega_{ij}, \tau_a) & \text{for } \tilde{a}_{ij} \in \tilde{\mathbf{A}} \text{ and } \omega_{ij} \in \mathbf{\Omega} \end{cases} \tag{4}$$

where $\tau \in (0, \infty)$ is a hyperparameter representing the temperature, $\omega_{ij} = \log(\theta_{ij}/(1-\theta_{ij}))$, $\eta_{ij} = \log \mathbf{q}_{ij}$, and $\xi_i = \log \mathbf{p}_i$. The reparameterization trick is also applied to make this distribution differentiable, facilitating gradient-based optimization. An independent random variable $\epsilon \sim \mathrm{Uniform}(0,1)$ to adjust the sampling function as follows:

$$\begin{cases} \tilde{x}_i \sim \mathrm{Softmax}((\xi_i - \log(-\log \epsilon))/\tau_z) \\ \tilde{z}_{ij} \sim \mathrm{Softmax}((\eta_{ij} - \log(-\log \epsilon))/\tau_z) \\ \tilde{a}_{ij} \sim \mathrm{sigmoid}((\omega_{ij} - \log \epsilon - \log(1-\epsilon))/\tau_a) \end{cases} \tag{5}$$

With these two modifications it's possible to draw samples that correspond to the distribution, without losing differentiability. Finally, the learning objective is approximated using the Monte Carlo method with K samples:

$$\max_{\Theta,Q,P} \mathbb{E}_{G \sim P(G)}[L(A,Z,X)] \approx \max_{\Omega,H,\Xi} \mathbb{E}_{\epsilon \sim U(0,1)}[L(\widetilde{A}, \widetilde{Z}, \widetilde{X})] \approx \max_{\Omega,H,\Xi} \frac{1}{K} \sum_{k=1}^{K} L(\widetilde{A}, \widetilde{Z}, \widetilde{X}). \tag{6}$$

where $\widetilde{A}, \widetilde{Z}, \widetilde{X}$ represent the continuously relaxed versions of $A, Z, X$ (described above) obtained after applying the reparameterization trick.

**Regularization.** L1 and L2 regularization are used to avoid gradient saturation and to also encourage model sparsity. Equation 7 shows how both types are applied to the $\mathbf{\Omega}$ parameter, which describes the categorical distribution for each connection in the adjacency matrix:

$$R_{L_1} = \|\mathbf{\Omega}\|_k \quad \text{with } k \in \{1,2\} \tag{7}$$

To limit the size of the explanation graphs and to prevent them from growing indefinitely with repeated discriminative patterns, a budget penalty regularization is also used. This is expressed in Equation 8 where the additional parameter B represents the expected maximum number of edges in the explanation graph.

$$R_{L_1} = \text{softplus}(||\text{sigmoid}(\mathbf{\Omega})||_1 - B)^2 \tag{8}$$

Lastly, as inspired by the work of Luo et al. (2020a), a connectivity incentive term is also applied to promote correlation, which is done by minimizing the Kullback-Leibler divergence $\text{D}_{\text{KL}}$ between the probabilities of each pair of edges that share a common node. This can be seen in Equation 9, where $P_{ij}$ represents the Bernoulli distribution parameterized by $\text{sigmoid}(\omega_{ij})$.

$$R_c = \sum_{i \in V} \sum_{j,k \in \mathcal{N}(i)} \text{D}_{\text{KL}}(P_{ij}||P_{ik}) \tag{9}$$

## 5 Methodology

The GNNInterpreter implementation, alongside with datasets, pre-trained classifiers and experiment notebooks is publicly available[1]. Consequently, we make use of the authors' code, in 2 different versions, namely the initial and the latest version, and with some minor bugfixes and improvements. We set up an environment that replicates the required versions for all frameworks and packages and run all experiments consistently in this environment. Additionally, we enhance and extend the work of Wang and Shen by conducting additional experiments and presenting further results on the reddit-binary dataset.

It's important to note that the GitHub implementation by the authors is not officially endorsed. While labeled 'official' and hosted on the main author's GitHub page, it is not directly referenced in the original paper. However, after extensively reviewing the code, we have found that it closely matches the architecture described in the paper. For that reason, we decided to use this implementation with only minor changes, as mentioned earlier, and addressing ambiguities where the paper lacked clarity (see Section 5.3 and Appendix E). We have made available the revised code we used for all experiments, together with demo notebooks[2].

### 5.1 Datasets

The experiments conducted in the original paper, both quantitative and qualitative, involve the use of 5 different datasets with the aim of demonstrating the flexibility of GNNInterpreter of working with various feature types.

**Synthetic Datasets.** The authors have created 3 synthetic datasets, namely Cyclicity, Motif and Shape, each with their unique characteristics:

- The **Shape** dataset consists of 5 classes of graphs: Lollipop, Wheel, Grid, Star and Others. Each class contains graphs with the corresponding shape, while the Others class has graphs with random topology.
- Within the **Motif** dataset, graphs are classified based on the presence of specific motifs, including House, House-X, Complete-4, and Complete-5, each corresponding to a distinct class label. Additionally, there is a separate class encompassing graphs without any unique motifs. Nodes in this dataset are characterized by a categorical feature with 5 potential color values.
- The graphs in the **Cyclicity** dataset are identified by edge features with 2 potential values: green or red. The classification involves 3 labels: Red-Cyclic, Green-Cyclic and Acyclic graphs. A graph belongs in the red/green cyclic classes only if it contains a cycle formed of exclusively red or green edges. Acyclic graphs or graphs that contain a heterogeneous edge cycle are categorized as Acyclic.

For further insights into each synthetic dataset generation procedure, please refer to Appendix F.

**Real-world Dataset.** Additionally, the authors used the real-world MUTAG dataset (Morris et al. (2020)). This dataset contains 188 graphs representing molecules with either the class label mutagenic or non-

---

[1]https://github.com/yolandalalala/GNNInterpreter
[2]https://github.com/MeneerTS/GNNinterpreter_Reproduction

mutagenic. These graphs have an average of 17.93 nodes and 19.79 edges. Both the nodes and the edges have labels (although edge labels are unused), representing atoms and types of chemical bonds respectively. Important to note is that hydrogen atoms were removed from all graphs . The original paper that collected the dataset (Debnath et al. (1991)) notes that the main factors for mutagenic properties were the hydrophobicity of the molecule, the energy of the lowest unoccupied electron orbital and the presence of three or more fused rings. For our purposes, as the hydrogen atoms have been removed, this translates to the presence of many NO2 bonds and the presence of many fused rings. However, it is essential to acknowledge that determining the mutagenicity of a molecule is a highly challenging task and that these assumptions give a simplified view that is only valid for the purposes of testing out the GNNInterpreter.

**Additional Dataset.** Since the original paper only tested the model on one real-world dataset, we have broaden the scope of our investigation for Claim 1 and 2 by using the Reddit-Binary dataset (Yanardag & Vishwanathan (2015)). This dataset comprises 2000 real-world examples of text posts and their corresponding comments, known as threads, from two different types of posts on Reddit. These posts are represented as graphs where nodes correspond to users and edges denote interactions, such as comments on each other's posts or replies to comments. The posts are categorized into 2 classes based on their origin within specific subreddits: the first class includes threads from the *IAmA* and *Askreddit* subreddits, characterized by question-answer interactions, while the second class consists of discussion threads from *TrollXChromosomes* and *atheism*, where multiple users engage in diverse interactions. The distribution of these classes is balanced, although the first class has larger graphs, with 641.24 nodes and 1471.90 edges on average, while the second class only has 217.99 nodes and 519.11 edges on average.

## 5.2 GNN architectures

The experimental study of the original paper employs different GNN classifiers trained on each of the datasets mentioned above. For MUTAG, Motif and Shape datasets, a GCN model is used (Kipf & Welling (2017)), consisting of 3 graph layers of width 64, a global mean pooling and 2 dense layers. For the additional Reddit-Binary dataset, this architecture remained the same but with 5 layers to accommodate the larger graph size. Lastly, the GNN classifier used for the Cyclicity dataset is a deep NNConv model (Gilmer et al. (2017)) with 5 NNConv layers of width 32, and again a global mean pooling layer and 2 dense layers at the end. Both model architectures use LeakyReLU activation.

## 5.3 Hyperparameters

To closely replicate the original experiments, we first adopted the same hyperparameter values as in the paper whenever they were explicitly mentioned. This entailed using $\tau = 0.2$ for the Concrete Distribution temperature, a sample size K = 10 for Monte Carlo samplings, an SGD optimizer with learning rate 1 during training, a max node count of 20 and an embedding similarity $\mu = 10$ for MUTAG and $\mu = 1$ for all other datasets. In our experimentation with regularization weights, we discovered that certain values led to empty graph generation errors. As we have noticed a discrepancy between the values mentioned in the paper and the ones used in the code notebooks provided in the original GitHub repository, the next step was to try this other set of values. However, for some datasets these values also led to sub optimal results. Please refer to Appendix A for the actual paper and notebook parameters, and to Appendix B for results using these values. Finally, we extensively tuned these weights separately for each dataset, in order to achieve the highest quantitative results, just as would be done when applying GNNInterpreter to a new problem. Note that adjustments were not necessary for all datasets, as for example, the paper parameters already have close to perfect quantitative results on the MUTAG dataset, while the notebook ones achieved comparable results on the Cyclicity dataset. The final values based on which we later report the results can be found in Appendix C.

For the additional Reddit-Binary dataset, hyperparameter tuning had to be done from scratch. Due to the larger size of the graphs, a larger max node count of 50 was required, although it was kept as small as possible to ensure readability of the explanations. The other hyperparameters can be found in Appendix C.

### 5.4 Experimental setup

**GNN reproduction.** To validate the accuracy of the base GNN models, we tested both the provided model checkpoints and retrained models. First, we loaded the provided checkpoints into the architectures outlined in section 5.2. To test these models, we used the same test-train split as reported in the authors' notebooks. Subsequently, we retrained the models according to the original paper specifications, employing Kaiming parameter initialization and an AdamW optimizer with a 0.01 learning rate and 0.01 weight decay.

**GNN Interpreter.** To verify the first three claims, we measured the performance of the GNNInterpreter both quantitatively and qualitatively like in the original paper. The quantitative analysis involved passing model-generated explanation graphs to the GNN classifiers to obtain class probabilities. The main rationale guiding the original authors' decision for this was that both GNNInterpreter and XGNN are designed to maximize target class scores. As such, a GNNInterpreter model is considered to have good quantitative performance if the explanation graphs yield high probabilities for the correct class. Based on this, the original study sampled 1000 explanation graphs from a single GNNInterpreter and averaged over the resulting GNN class probabilities to measure the quantitative performance.

During our reproduction however, we noticed that the performance of a GNNInterpreter depends heavily on the random initialization and thus we decided to train 100 different models for each class with seeds between 0 and 100. We then applied the same 1000 explanation graph quantitative analysis for each model individually and averaged over all models. Furthermore, we quantified the frequency with which the GNN Interpreter could be categorized as a good model, defined by a correct class probability of at least 0.9. In contrast, we defined a bad model as either a model that generated empty graphs or graphs with target class probability less than 0.1. Building upon these definitions, we analyze and report the quantitative analysis for each class in all datasets using the worst and best performing models, as well as by averaging over all 100 models. To test Claim 4, we used the same 100 models to get the average training times per class for GNNInterpreter across all 4 datasets and for XGNN with the MUTAG dataset.

For the qualitative analysis, we used explanation graphs that got target class probability of 1.0 whenever possible, and otherwise explanations generated by the best model out of 100 seeds. We then performed visual inspections on these graphs, paying particular attention to properties mentioned in the original paper.

We did some further analysis on the training dynamics after encountering instability during training. We tracked the explanation graph size and correct class probability throughout training iterations. We used the Motif, Cyclicity and Reddit-Binary datasets for the analysis. Reddit-Binary was chosen because of its significantly larger graphs. Motif and Cyclicity were chosen because they were two datasets that we encountered the most instability while still being able to get good results occasionally.

**Additional Dataset.** Due to the larger graph and model size of the Reddit-Binary dataset, replicating the same experiments as with the other datasets was computationally infeasible. Therefore, we only trained the base GNN once and averaged GNNInterpreter results over just 10 seeds. In addition, we constructed a random baseline to examine the effect of the substantial difference in graph size between the two classes.

**Ablation study.** The ablation study reported in the original paper aimed to illustrate the importance of the cosine similarity term in the learning objective for generating meaningful explanations. This was done by setting the $\mu$ parameter to 0 in Equation 1. To recreate this experiment, we trained a GNNInterpreter on the mutagen class, using the same parameters as reported in the original paper and, once more, averaged over 5 different seeds for a more accurate representation.

**Verification study.** The verification study performed by the original authors aimed to confirm that the GNNInterpreter's explanation graphs were actually explaining the behavior of the GNN classifier. As part of the qualitative analysis on the Motif dataset, rules that represent the discriminative features the GNN uses were manually extracted from the explanation graphs. These rules were then used to create 8 new motifs that share common features but are different from the ground truth motif. 5000 base graphs were generated and attached to each of the 8 fake motifs and the ground truth motif. These new data-points were passed through the GNN model. If the extracted rules and explanation graphs reflect the GNN faithfully, it is expected for the GNN to misclassify the 8 new fake motifs as the ground truth motif. This will prove that

the explanation graph generated by GNNInterpreter included features that are actually used by the GNN to classify the ground truth class.

## 5.5 Computational requirements

All our experiments were run using a single CPU, an AMD Ryzen 5900x. We made an effort to replicate the Python environment as closely as possible to the one specified by the authors. As such, we used Python 3.9.0 and PyTorch 2.0.0. For PyTorch Geometric, we employed version 2.3.0, as indicated in the authors' GitHub, rather than the version specified in the paper. We opted for this approach because we ran all experiments using the code from this source. In addition, we used the libraries torch-scatter and torch-sparse for torch 2.0.0 with cuda version 118, although no GPU is required. This environment can be found in our repository. Prior to use, please ensure to follow the provided installation guide. All models, including XGNN, use 83 watts of power during training. The total training time for averaging over 100 seeds, experimenting on GNNInterpreter, and training the GNNs was 15.26 hours, with a total power consumption of 1.267 kWh.

# 6 Results

## 6.1 GNN accuracies

Our analysis on both the pretrained and retrained GNN models revealed that for all datasets except for MU-TAG, both approaches consistently yielded lower, yet still relatively comparable, accuracy scores compared to those reported in the paper. Averaged results can be found in Table 1. Of note is that even though the average performance on the MUTAG dataset was significantly lower, some individual models perform about as well as reported in the original paper, reaching 0.944 accuracy in our limited test of 5 seeds (reflected by the high variance). Performance similar to the original paper can thus be reached by finding the right seed.

| Dataset | Pre-trained model | Re-trained model | Original paper accuracy |
|---|---|---|---|
| MUTAG | 0.8333 | $0.7222 \pm 0.1267$ | 0.9468 |
| Cyclicity | 0.9493 | $0.9302 \pm 0.0091$ | 0.9921 |
| Motif | 0.9966 | $0.991 \pm 0.00235$ | 0.9964 |
| Shape | 0.9812 | $0.9795 \pm 0.0016$ | 0.9725 |
| REDDIT-BINARY | - | 0.8650 | - |

Table 1: The accuracies of the GNN models. Retraining was done on 5 different seeds, for which the mean and standard deviation are reported. The GNN for the Reddit-Binary dataset was only trained once due to its long training time.

## 6.2 Quantitative and qualitative evaluation

As explained in section 5.4, we performed both a quantitative and qualitative analysis. Quantitative results, in the form of predicted class probabilities and training times, are presented in Table 2. One explanation graph per class generated from the best interpreter that was found can be seen in Table 3, while more qualitative results are later presented in Appendix D.

**MUTAG.** The quantitative analysis shows that training the GNNInterpreter is almost always successful in terms of reaching high class probabilities. However, the qualitative results do not seem to match with the examples from the dataset and for the mutagen class they also do not match with the result from the original paper. In terms of training times for this dataset, we observe a significant disparity between GNNInterpreter and XGNN. Specifically, the average training time for the former is 0.79, while for the latter, it is 38.83.

**Cyclicity.** The quantitative results show that the model is capable of achieving a perfect predicted probability for all 3 classes in this synthetic dataset, as it can be seen in the column for the Best model in Table 2. The qualitative results further corroborate this, with the explanation graphs containing a single red and green cycle for the first 2 classes respectively, and a cycle with heterogeneous edge features for the Acyclic class. However, it is crucial to emphasize that under different random initialization, the model predominantly predicts correct results only the Red Cyclic class, while for the other 2 classes the likelihood of a good model is only 0.22 and 0.37 respectively.

**Shape.** Quantitative results on the Shape dataset revealed that GNNInterpreter struggled to reliably converge, except for the star class. While this class achieved an average accuracy of 1, other classes did not perform as well. The lollipop class has a low accuracy of 0.22, while the wheel class had 0.84 and the grid class 0.78.

| | | Average of all Models | Best Model | Worst Model | Percentage of good models | Percentage of bad models | Training time (s) |
|---|---|---|---|---|---|---|---|
| MUTAG | Mutagen | 0.987 ± 0.100 | - | - | - | - | 38.83 |
| (XGNN) | Nonmutagen | 0.999 ± 0.002 | - | - | - | - | |
| MUTAG | Mutagen | 0.999 ± 0.006 | 1.000 ± 0.000 | 0.921 ± 0.254 | 1.00 | 0.00 | 0.79 |
| (GNNInterpreter) | Nonmutagen | 0.943 ± 0.068 | 1.000 ± 0.000 | 0.330 ± 0.429 | 0.87 | 0.00 | |
| Cyclicity | Red Cyclic | 0.926 ± 0.0677 | 1.000 ± 0.000 | 0.000 ± 0.000 | 0.84 | 0.02 | 24.85 |
| (GNNInterpreter) | Green Cyclic | 0.665 ± 0.372 | 1.000 ± 0.000 | 0.101 ± 0.290 | 0.22 | 0 | |
| | Acyclic | 0.525 ± 0.120 | 1.000 ± 0.000 | 0.000 ± 0.000 | 0.37 | 0.40 | |
| | House | 0.787 ± 0.220 | 0.991 ± 0.006 | 0.000 ± 0.000 | 0.41 | 0.08 | 19.17 |
| Motif | House-X | 0.276 ± 0.085 | 0.999 ± 0.009 | 0.000 ± 0.000 | 0.11 | 0.63 | |
| (GNNInterpreter) | Complete-4 | 0.077 ± 0.020 | 0.995 ± 0.052 | 0.000 ± 0.000 | 0.06 | 0.91 | |
| | Complete-5 | 0.131 ± 0.034 | 0.997 ± 0.053 | 0.000 ± 0.000 | 0.07 | 0.82 | |
| | Lollipop | 0.222 ± 0.294 | 0.43 ± 0.374 | 0.096 ± 0.199 | 0.00 | 0.01 | 23.48 |
| Shape | Wheel | 0.84 ± 0.279 | 0.997 ± 0.056 | 0.058 ± 0.231 | 0.45 | 0.02 | |
| (GNNInterpreter) | Grid | 0.782 ± 0.327 | .911 ± 0.216 | 0.612 ± 0.408 | 0.02 | 0.00 | |
| | Star | 1.000 ± 0.001 | 1.000 ± 0.000 | 0.987 ± 0.109 | 1.00 | 0.00 | |
| Reddit-Binary | Question-Answer | 0.8454 ± 0.019 | 0.89199 | 0.72159 | - | - | 25.774 |
| (GNNInterpreter) | Discussion | 0.989 ± 0.000 | 0.9889 | 0.9889 | - | - | |

Table 2: The quantitative results for the 4 original datasets plus and the Reddit-Binary dataset. The metric used per model is the average class probability of 1000 explanation graphs. Reported is the average of the class probabilities for 100 models across different seeds, with the exception of Reddit-Binary where testing was done only using 10 seeds. We also report the probabilities generated by the best and worst interpreter, together with the percentages of good and bad models obtained. The standard deviations represent the standard deviation obtained from each quantitative test, averaged over all models.

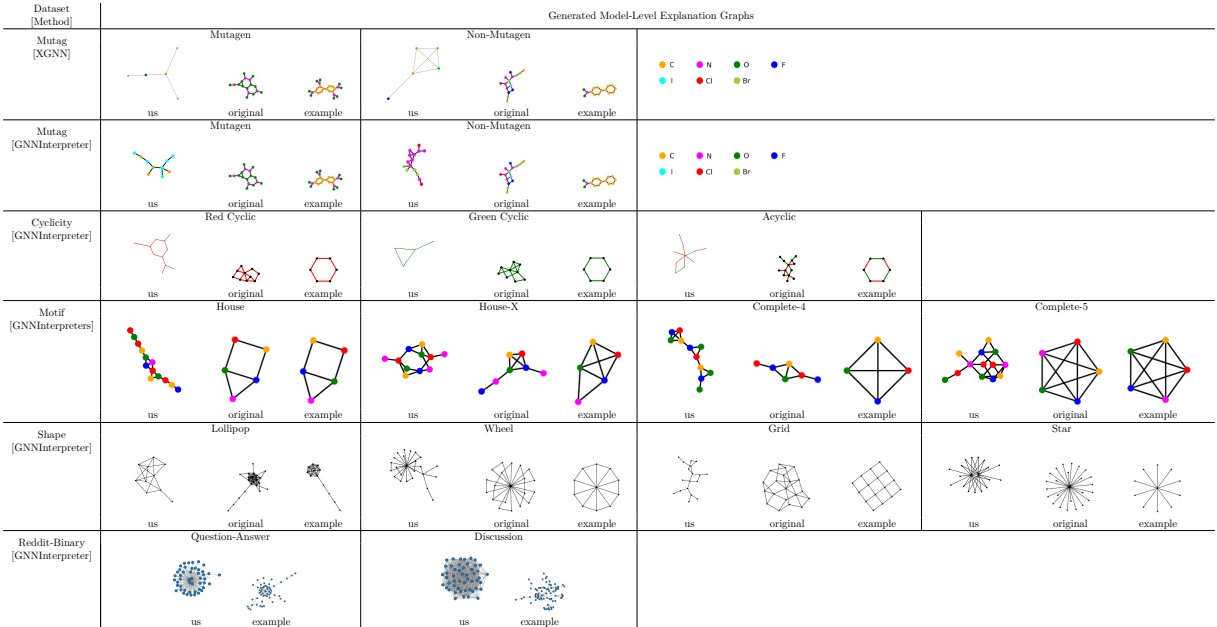

Table 3: The qualitative results for the 4 datasets. For easy comparison, 3 figures for each class in all datasets are reported, namely (1) an explanation graph generated while reproducing the experiments, (2) the explanation graph reported in the original paper and (3) an example graph from the dataset, chosen to have the highest class probability that was found for its respective target class.

**Motif.** The quantitative results of the Motif dataset seen in Table 2 show that GNNInterpreter's performance decreases as the motif becomes less unique. Complete-4 and Complete-5 motifs are much more likely to occur in random graphs than House and House-x. Furthermore, Complete-4 and Complete-5 classes had a good model only 6% and 7% of the time. Despite the rarity of good models, we managed to get at least 1 model for each class that achieved target class probability 1. We believe the high variance of our results may stem from the instability during training which we further explored in 6.4.

As for qualitative analysis we can see in Table 3 that if a GNNInterpreter with high target class probabilities is used, we can get the motif, or part of the motif in the explanation graphs. Small triangles resembling the roof of the house, x shaped connection of House-x, and many interconnected nodes like complete-5, are key features of the ground truth motifs present in the respective explanation graphs. The explanation graph for Complete-4 actually contains the entire ground truth motif positioned at the top left. We also observed an exact match with the house-x motif which is the first image of house-x row in Appendix Table14. However, this graph achieved a class probability of 0.62 instead of 1.

**Verification study.** In their verification study, the original authors derived a discrete rule from the explanation graphs through qualitative analysis. However, our explanation graphs, despite having a class probability of 1.0, were significantly larger and lacked the same level of interpretability as those in the original paper. This made qualitative rule extraction impractical, thus rendering the replication of this experiment unfeasible.

**Ablation study.** The explanation graphs generated did not change when changing $\mu$ to zero for 19 out of 20 of the seeds checked. The logits also stayed exactly the same, except for one case where it changed. In that case the GNNInterpreter did not manage to converge on the correct class and the logit was actually negative with $\mu = 0$ and positive with $\mu = 10$. This means that the impact of changing mu is minimal for this model and dataset, only changing the outcome in some cases.

### 6.3 Results beyond original paper

**Reddit-Binary.** The random graph baseline results indicate that when the maximum node count is set to 50, the Discussion class is consistently selected by default, with an average class probability of 0.9887. Moreover, analysis of the Reddit-binary dataset, as presented in Tables 2 and 3, illustrates that the GNNInterpreter achieves both good quantitative and qualitative results for the Ask-question class. This suggests that the model effectively captures a distinguishing feature of this class, namely the presence of a few experts answering numerous questions. Despite its robust quantitative performance, GNNInterpreter appears to generate nearly random graphs for the Discussion class, providing limited explanation. This outcome is however somewhat expected, given that random graphs perform well for this class, as evident from the baseline.

**MUTAG.** Given the discrepancies observed between the quantitative and qualitative analyses of GNNInterpreter's performance on the MUTAG dataset, particularly the lack of alignment with dataset examples, we sought to investigate whether this issue steams from the inherent limitations of the dataset or within the GNNInterpreter model itself. To address this, we conducted a comparative analysis with GNNExplainer (Ying et al. (2019a)), and we report the obtained qualitative results in Table 5. We chose GNNExplainer due to its resemblance to GNNInterpreter in implementing a budget penalty for generating compact explanations, along with the straightforward experimental setup on this dataset. It is evident from these finding that GNNExplainer outperforms GNNInterpreter in terms of faithfulness and realism. Despite imperfections, we note that both graphs depict carbon rings accurately, with the graph for the mutagen class further featuring a NO2 bond.

### 6.4 Analysis of training instability

We observed the expected behavior of decreasing graph size and increasing correct class probability only with the Reddit-Binary dataset question-answering class as can be seen from Table 6.

| Dataset [method] | Generated Model-Level Explanation Graphs | | |
|---|---|---|---|
| Mutag [GNNExplainer] | Mutagen | Non-Mutagen | 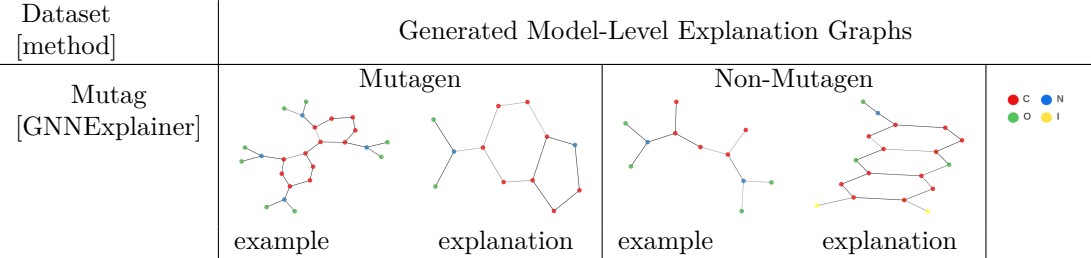 |
| | example          explanation | example          explanation | |

Table 5: Qualitative results on MUTAG dataset from GNNExplainer model.

Table 6: Graph Size and Correct Class Probabilities vs Iteration graphs of select examples that demonstrate different types of training behavior. Starting from top left: Expected behavior, Never-converging, Convergence, and Non-convergence.

Depending on the random seed, some initializations resulted in 0 class probability explanation graphs after 2000 iterations such as Motif House-x Seed 0, while other seeds managed to reach the early stopping criteria of at least 0.9 correct class probability like Motif House-x Seed 2, as seen in Table 6. We will refer to getting stuck at 0 probability for the entire training as the never-converging scenario. Reaching early stopping will be

referred as the convergence. There were also cases were GNNInterpreter learned a 0 correct class probability explanation graph even though it managed to get larger but higher probability explanation graphs during training. For example, Cyclicity Acyclic plot from Table 6, missed the local minima that would coincide with larger but higher probability explanation graphs. We will call this the non-convergence. These were the 4 main phenomena that we observed during our experiments. Certain classes were predisposed to experience a certain scenario based on their individual properties, yet these scenarios varied depending on the seed.

## 7 Discussion

### 7.1 Discussion of the results

This study assessed four claims regarding the effectiveness of GNNInterpreter as an explainability technique for Graph Neural Networks. Firstly, the GNNs used were reproduced to ensure a solid base for further analysis and they were found to match the original paper to a satisfying degree.

Regarding the realism and faithfulness of generated explanations, as per Claim 1, we have found that adjusting hyperparameter values yielded mixed quantitative and qualitative results across datasets, with a significant sensitivity to seed variation. While the technique holds promise for generating realistic results, its unreliability poses a significant challenge to achieving those in practice. Using the available code repository, the necessity of tuning numerous hyperparameters across various seeds undermines the usefulness of this technique. For some classes, like Lollipop from the Shape dataset, GNNInterpreter was never able to reliably achieve realistic results even after checking 100 seeds. This indicates its performance is also heavily influenced by the dataset and class.

Upon our analysis of training instability, we found the main reason to be the graph size of the dataset. Even though we use relaxation to model the graphs as continuous distributions, the loss behaves discretely when the graph size is small. The class probabilities and logits we get from the GNN classifiers vary drastically since removing even one edge from the graph can drastically change the probability from 1 to 0. For example, Acyclic class from the Cyclicity class has a discrete loss signal from the GNN since removing a single edge can make it Acyclic or adding a single edge can make it cyclic. This problem is not encountered in datasets like Reddit-Binary that have large graphs. Impact of a single edge is much lower on the GNN's decision in Reddit-Binary compared to Motif or Cyclicity datasets.

Another reason we think could explain the instability in GNNInterpreter's training procedure is the implementation of the dynamic weighting of the budget penalty, combined with the arbitrary selection of an early stopping criterion. These were design decisions not mentioned in the original paper and during testing we observed them to have a large impact on model performance. Sometimes their presence was the key to achieving convergence, while at other times they lead to non-convergence. Both dynamic budget weight penalty and early stopping depended on the max number of nodes parameter. The weight of the budget penalty was increased if the explanation graph was larger than the max number of nodes and had a correct class probability higher than a certain threshold. If the probability threshold was not reached, the budget penalty weight was decreased to incentivize exploration. The max number of nodes parameter was not mentioned in the original paper, and needs to be finetuned for each dataset due to variations in graph sizes. Our experiments on modifying and removing these all together can be found in Appendix E. A simple way to reduce non-convergence would be to save the best model during training and remove the early stopping.

The final type of training instability, is the never-converging scenario. We observed this scenario mostly in classes with highly specific structures such as complete-4 and complete-5 from Motif dataset. If the sampled graphs are not large enough, it is unlikely that 4 or 5 nodes all completely connected will occur. It is very unlikely that GNNInterpreter will explore larger graphs because of the budget penalty. For complicated or highly specific classes, a large graph must be sampled to decrease the dependence on random initialization. Sampling larger graphs slows down GNNInterpreter significantly. Also the size of the sampled graph is introduced as a new parameter not mentioned in the original paper. Determining both the size of the sampled graph and the maximum number of nodes for dynamic weighting necessitates domain knowledge for appropriate selection. A max size of a class parameter could be used to avoid using domain knowledge and ensuring a large enough graph to sample from.

As for the faithfulness of explanation graphs, we found that, for the MUTAG dataset, good quantitative results did not lead to good qualitative results. This indicates that while using this technique on a new problem, it is possible that the provided explanations are not representative even after proper fine-tuning. GNNInterpreter yielded faithful qualitative results for Reddit-Binary highlighting the importance of having a large enough graph to ensure the class probability loss is easily optimizable. Overall, using the only publicly available code repository, we found that this method requires domain knowledge for tuning parameters, and is unstable with datasets that have small graphs. This shows that Claim 1 holds for datasets with large graphs where the impact of a single node on the classification outcome is relatively small. Claim 1 does not hold for datasets with smaller graphs. For these datasets, faithful explanations can reliably generated only by extensive parameter tuning with domain knowledge or trying random initializations.

Moreover, by being able to train at least some good interpreters on multiple datasets, each with unique types of node features, our results provide evidence in support of Claim 2. While results were not good for the Shape dataset, the inclusion of the Reddit-Binary dataset shows that GNNInterpreter can perform well without node or edge features. The Cyclicity dataset showed good results for datasets with only edge features, while MUTAG and Motif showed promising results for datasets with just node features. This just leaves the case of a dataset with both node and edge features, which could be explored further in future research. However, our results do not indicate that GNNInterpreter would encounter challenges with such a dataset.

Despite the limitation of solely experimenting with XGNN on the MUTAG dataset, our results provide sufficient basis for drawing meaningful conclusions about Claim 3 and 4. As evident from our qualitative results, GNNInterpreter did not produce explanation graphs aligned significantly better with the dataset for either the mutagen or the non-mutagen class, compared to XGNN. Furthermore, based on our qualitative analysis, explanations generated by GNNExplainer outperformed GNNInterpreter. Therefore, for the MU-TAG dataset our results contradict Claim 3. While we were able to replicate the training times for a single model as reported by the authors, it is worth noting that for most datasets, we had to train on multiple seeds (as many as 20) to obtain a single good interpreter. Particularly for classes with greater complexity, we observed a decrease in the number of good models. As such, we believe that the training times should better reflect the actual amount of time required to achieve a good model either by trying random seeds or tuning parameters and sampling larger graphs for complicated classes. Nevertheless, for a single model, we have found GNNInterpreter to be nearly 38 times faster than XGNN and we thus affirm the validity of Claim 4 on the MUTAG dataset.

GNNInterpreter can be improved to be more reliable, stable, and require less domain knowledge at the cost of increased training times. As mentioned in the discussion, using max class size for sampled graph size can eliminate the never-converging problem, reducing reliance on random initializations as well as domain knowledge. Exploration-exploitation trade off can be better balanced by carefully selecting max number of nodes, and saving the best model.

## 7.2 Reflection: What was easy? What was difficult?

The architecture of GNNInterpreter was clearly described in the paper, both intuitively and formally. The set-up and objectives of each experiment were also explicitly mentioned with enough level of detail. Despite not being linked to the paper, the original code implementation was publicly available on one of the author's personal GitHub. Beyond that, model checkpoints and the code for generating the synthetic datasets was also available together with some general experimental notebooks.

However, it was not trivial to recreate the experiments. The main reasons for that include the perplexing structure of the code base, the lack of documentation, the abundance of unused or erroneous code, but more importantly the numerous discrepancies we found regarding implementation details and hyperparameters values between the paper and the open-source implementation.

### 7.3 Communication with original authors

We have contacted the original authors of the paper seeking clarification on various aspects of the paper and the associated code. Specifically, we included topics like confirming the official repository and recent updates, identifying potential bugs in the code base, understanding the configuration of regularization weights, confirming seed usage and averaging in reporting results as well as resolving mismatches between hyperparameter values in the appendix and code repository. Moreover, we have also made inquiries about certain aspects that were present in the code but not mentioned in the paper, such as thresholding qualitative results and adding a mean penalty to the weighting criterion. Unfortunately, we have not received any response.

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

## 8 Appendix

## A Original hyperparameter values

Table 7 gives the regularization weights reported in the original paper, while Table 8 gives the values from the experimental notebooks found in the authors' GitHub repository.

| Dataset | Class | Regularization Weights | | | |
|---|---|---|---|---|---|
| | | $R_{L_1}$ | $R_{L_2}$ | $R_b$ | $R_c$ |
| MUTAG | Mutagen | 10 | 5 | 20 | 1 |
| | Nonmutagen | 5 | 2 | 10 | 2 |
| Cyclicity | Red Cyclic | 10 | 5 | 10000 | 100 |
| | Green Cyclic | 10 | 5 | 2000 | 50 |
| | Acyclic | 10 | 2 | 5000 | 50 |
| Motif | House | 1 | 1 | 5000 | 0 |
| | House-X | 5 | 2 | 2000 | 0 |
| | Complete-4 | 10 | 5 | 10000 | 1 |
| | Complete-5 | 10 | 5 | 10000 | 5 |
| Shape | Lollipop | 5 | 5 | 1 | 5 |
| | Wheel | 10 | 5 | 10 | 0 |
| | Grid | 1 | 1 | 2 | 0 |
| | Star | 10 | 2 | 200 | 0 |

Table 7: Values mentioned in the original paper for the regularization weights of GNNInterpreter.

| Dataset | Class | Regularization Weights | | | |
|---|---|---|---|---|---|
| | | $R_{L_1}$ | $R_{L_2}$ | $R_b$ | $R_c$ |
| MUTAG | Mutagen | 1 | 1 | 1 | 0 |
| | Nonmutagen | 1 | 1 | 1 | 0 |
| Cyclicity | Red Cyclic | 2 | 2 | 1 | 5 |
| | Green Cyclic | 2 | 2 | 1 | 5 |
| | Acyclic | 2 | 2 | 1 | 5 |
| Motif | House | 1 | 1 | 1 | 0 |
| | House-X | 1 | 1 | 1 | 0 |
| | Complete-4 | 1 | 1 | 1 | 0 |
| | Complete-5 | 1 | 1 | 1 | 0 |
| Shape | Lollipop | 1 | 1 | 0 | 15 |
| | Wheel | 1 | 1 | 0 | 10 |
| | Grid | 1 | 1 | 0 | 20 |
| | Star | 1 | 1 | 0 | 10 |

Table 8: Values found in the authors' original experimental notebooks for the regularization weights of GNNInterpreter.

## B Tests on Provided Hyperparameters

Table 9 and Table 10 below display a small-scale version of our experimental results averaged over 5 random seeds, where each dataset has been tested using the paper and the notebook parameters respectively. Important to note is that we didn't perform the same complete analysis with reporting good and bad models, the purpose of this set-up being solely to motivate the necessity of the hyperparameter tuning for obtaining the main results. The paper parameters result in empty graphs for Motif dataset.

| Dataset | Predicted Class Probability by GNN | | | |
|---------|----------------------------------|---|---|---|
| MUTAG | Mutagen: $1.000 \pm 0.000$ | Nonmutagen: $0.9990 \pm 0.002$ | | |
| Cyclicity | Red Cyclic: $0.200 \pm 0.000$ | Green Cyclic: $0.580 \pm 0.056$ | Acyclic: $0.730 \pm 0.095$ | |
| Motif | House: $0 \pm 0.000$ | House-X: $0.000 \pm 0.000$ | Complete-4: $0.000 \pm 0.000$ | Complete-5: $0.000 \pm 0.000$ |
| Shape | Lollipop: $0.000 \pm 0.000$ | Wheel: $0.000 \pm 0.000$ | Grid: $0.000 \pm 0.000$ | Star: $0.000 \pm 0.000$ |

Table 9: The reproduced quantitative results for the 4 datasets using the hyperparameters in the original paper, averaged over 5 different seeds.

| Dataset | Predicted Class Probability by GNN | | | |
|---------|----------------------------------|---|---|---|
| MUTAG | Mutagen: $1.000 \pm 0.000$ | Nonmutagen: $0.9712 \pm 0.058$ | | |
| Cyclicity | Red Cyclic: $1.000 \pm 0.000$ | Green Cyclic: $0.490 \pm 0.476$ | Acyclic: $0.447 \pm 0.113$ | |
| Motif | House: $0.721 \pm 0.195$ | House-X: $0.181 \pm 0.048$ | Complete-4: $0.347 \pm 0.096$ | Complete-5: $0.225 \pm 0.028$ |
| Shape | Lollipop: $0.7935 \pm 0.053$ | Wheel: $0.8965 \pm 0.27$ | Grid: $0.0881 \pm 0.0988$ | Star: $0.9806 \pm 0.14$ |

Table 10: The reproduced quantitative results for the 4 datasets using the hyperparameters in the original notebooks, averaged over 5 different seeds.

## C  Tuned hyperparameter values

Table 11 reports the regularization weights used for the quantitative and qualitative evaluations presented in the body of this paper.

| Dataset | Class | Regularization Weights | | | |
|---------|-------|-------|-------|-------|-------|
| | | $R_{L_1}$ | $R_{L_2}$ | $R_b$ | $R_c$ |
| MUTAG | Mutagen | 10 | 5 | 20 | 1 |
| | Nonmutagen | 5 | 2 | 10 | 2 |
| Cyclicity | Red Cyclic | 2 | 2 | 1 | 5 |
| | Green Cyclic | 2 | 2 | 1 | 5 |
| | Acyclic | 2 | 2 | 1 | 5 |
| Motif | House | 1 | 1 | 1 | 0 |
| | House-X | 1 | 1 | 1 | 0 |
| | Complete-4 | 1 | 1 | 1 | 0 |
| | Complete-5 | 1 | 1 | 1 | 0 |
| Shape | Lollipop | 1 | 1 | 0 | 15 |
| | Wheel | 1 | 1 | 0 | 10 |
| | Grid | 1 | 1 | 0 | 50 |
| | Star | 1 | 1 | 0 | 20 |

Table 11: Refined regularization weights of GNNInterpreter for explaining the GNN models corresponding to each dataset.

## D    Multiple explanation graphs per class per dataset

Table 14 below augments the qualitative results reported in the paper with a few extra explanation graphs that were generated during the reproducibility study for each class in each dataset.

| Dataset | Class | Explanation Graphs |
|---------|-------|--------------------|
| MUTAG | Mutagen |  |
|  | Non-mutagen |  |
| Cyclicity | Red Cyclic |  |
|  | Green Cyclic |  |
|  | Acyclic |  |
| Motif | House |  |
|  | House-X |  |
|  | Complete-4 |  |
|  | Complete-5 |  |

Continued on next page

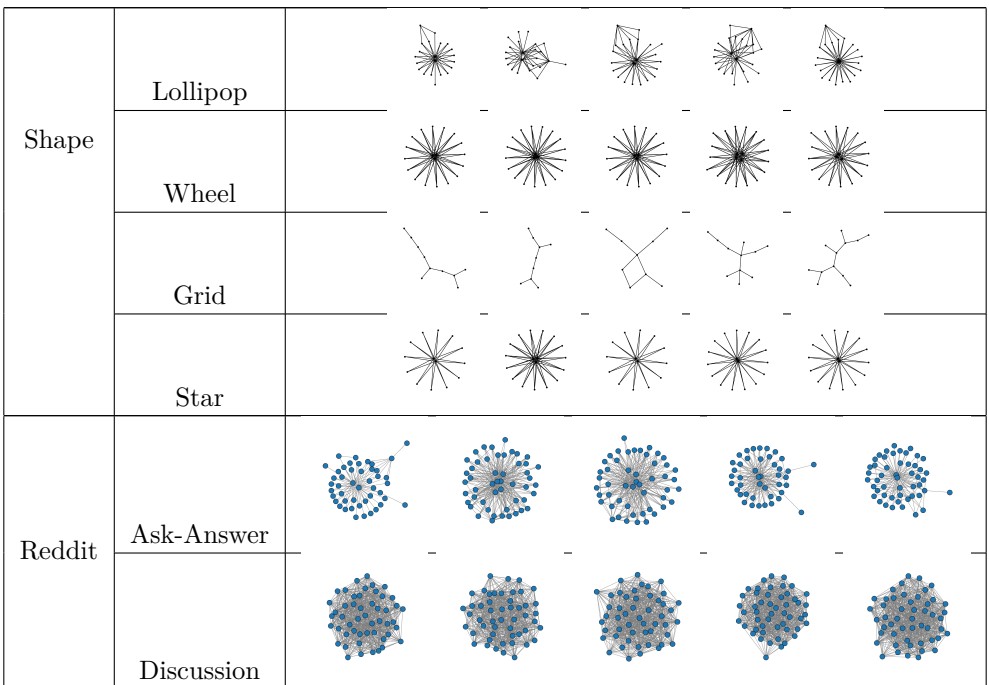

Table 14: Multiple qualitative results for all datasets.

# E   Dynamic Weight of Budget Penalty and Early Stopping

The official code repository of the original paper dynamically adjusts the weight of the budget penalty regularization during training, a detail overlooked in the original paper. The weight of the budget penalty decreases until the model achieves a target class probability of 0.9, after which it begins to increase again. This mechanism aims to enlarge the graph until the desired class probability is achieved, and subsequently reduce its size to produce a more concise explanation graph. The 0.9 target class probability criteria was determined by the original authors in the official code repository, which is employed alongside a maximum number of nodes criteria for early stopping purposes.

The official code repository featured two distinct implementations for dynamically adjusting the weight of the budget penalty. In the most recent commit, the weight was increased by multiplying it by 1.1 and decreased by multiplying it by 0.95 at each iteration, with the budget penalty weight initialized to 1. In contrast, the earliest commit in the repository modified the dynamic weight by adding and subtracting 0.1 from the budget penalty weight per iteration. In this latter version, the initial budget penalty weight could be adjusted via a parameter.

We experimented with the two versions of the repository, as well as with removing the dynamic weight penalty and early stopping criteria entirely. Results varied across datasets: Mutag showed similar class probabilities and graph sizes to the original paper with both versions, while the Cyclicity dataset performed better on the latest version, yielding smaller graphs compared to the oldest version. Conversely, Motif and Shape datasets performed significantly better by getting smaller explanation graphs on the oldest version of the repository. For example, Motif dataset gets explanation graphs of size 80 to 100 in the latest version, while the oldest version resulted in graphs of size 20. Size 20 is still higher than the results of the original paper, but it is much closer and the smallest size we can achieve.

Finally, we tried removing the dynamic weight penalty and the early stopping criteria all together to prevent the oscillation problem we mentioned in the discussion section. Instead of using the model from the latest iteration, we started using the model with the minimum loss. This change decreased effect the unstable

training had on our results. However, we got much larger explanation graphs of size 100 for Motif dataset, when we removed the dynamic weight which led us to not use this version.

## F  Synthetic dataset generation

The synthetic datasets used in this reproducibility study were generated following the identical procedures outlined in the original paper. These procedures are detailed for each dataset in Figure 1 below, using the algorithmic pseudocodes extracted from the original paper.

---

**Algorithm 3:** Cyclicity dataset generation procedure

**Edge Classes:** {RED, GREEN}
**Graph Classes:** {RED-CYCLIC, GREEN-CYCLIC, ACYCLIC}
**Input:** Rome Dataset
**Output:** A collection of pairs consisted of a graph and its label
1  **for** $G$ in Rome Dataset **do**
2      Randomly label each edge in $G$ as RED or GREEN.
3      **if** $G$ is acyclic **then**
4          **yield** $(G, \text{ACYCLIC})$
5          **break**
6      **while** more than one cycle exists in $G$ **do**
7          Remove a random edge from a random cycle in $G$.
8      Randomly pick a color $C_{\text{cycle}} \in \{\text{RED, GREEN}\}$.
9      Relable all edges in the remaining cycle in $G$ as $C_{\text{cycle}}$.
10     Randomly pick a color $C_{\text{edge}} \in \{\text{RED, GREEN}\}$.
11     Relable a random edge in the remaining cycle in $G$ as $C_{\text{edge}}$.
12     **if** $C_{\text{cycle}} = C_{\text{edge}} = \text{RED}$ **then**
13         **yield** $(G, \text{RED-CYCLIC})$
14     **else if** $C_{\text{cycle}} = C_{\text{edge}} = \text{GREEN}$ **then**
15         **yield** $(G, \text{GREEN-CYCLIC})$
16     **else**
17         **yield** $(G, \text{ACYCLIC})$

---

**Algorithm 4:** Motif dataset generation procedure

**Node Classes:** {RED, GREEN, ORANGE, BLUE, MAGENTA}
**Graph Classes:** {HOUSE, HOUSE-X, COMPLETE-4, COMPLETE-5, OTHERS}
**Input:** Rome Dataset
**Output:** A collection of pairs consisting of a graph and its label
1  Let $\{G_{\text{HOUSE}}, G_{\text{HOUSE-X}}, G_{\text{COMPLETE-4}}, G_{\text{COMPLETE-5}}\}$ be 4 motifs for the corresponding classes.
2  Define $\bigoplus$ as a graph operator such that $G_1 \bigoplus G_2$ generates a union graph $G_1 \cup G_2$ with an additional edge between a random node in $G_1$ and a random node in $G_2$.
3  **for** $G$ in Rome Dataset **do**
4      Randomly assign a color $C_{\text{node}} \in \{\text{RED, GREEN, ORANGE, BLUE, MAGENTA}\}$ for each node in $G$
5      Randomly select a label $L \in \{\text{OTHERS, HOUSE, HOUSE-X, COMPLETE-4, COMPLETE-5}\}$.
6      **if** $L = \text{OTHERS}$ **then**
7          Let $G_{\text{OTHERS}}$ be a random motif in $\{G_{\text{HOUSE}}, G_{\text{HOUSE-X}}, G_{\text{COMPLETE-4}}, G_{\text{COMPLETE-5}}\}$ but with a random edge being removed.
8      **yield** $(G \bigoplus G_L, L)$

---

**Algorithm 5:** Shape dataset generation procedure

**Graph Classes:** {LOLLIPOP, WHEEL, GRID, STAR, OTHERS}
**Output:** A collection of pairs consisted of a graph and its label
1  **for** *8000* times **do**
2      Randomly select a label $L \in \{\text{OTHERS, LOLLIPOP, WHEEL, GRID, STAR}\}$.
3      **switch** $L$ **do**
4          **case** LOLLIPOP **do**
5              Sample lollipop graph $G_{\text{LOLLIPOP}}$ with random number of head nodes $n \in \{4, ..., 16\}$ and random number of tail nodes $m \in \{4, ..., 16\}$.
6          **case** WHEEL **do**
7              Sample wheel graph $G_{\text{WHEEL}}$ with random number of non-center nodes $n \in \{4, ..., 64\}$.
8          **case** GRID **do**
9              Sample grid graph $G_{\text{GRID}}$ with random width $w \in \{2, ..., 8\}$ and random height $h \in \{2, ..., 8\}$.
10         **case** STAR **do**
11             Sample star graph $G_{\text{STAR}}$ with random number of non-center nodes $n \in \{4, ..., 64\}$.
12         **case** OTHERS **do**
13             Sample Binomial random graph $G_{\text{OTHERS}}$ with random number of nodes $n \in \{8, ..., 32\}$ and random edge probability $p \in [0.2, 1]$.
14     Add random number of noisy edges to $G_L$ with a random ratio $p \in [0, 0.2]$.
15     **yield** $(G_L, L)$

---

Figure 1: Synthetic datasets generation procedures.

