# OpenReview forum: "[Re] GNNInterpreter: A probabilistic generative model-level explanation for Graph Neural Networks"
_TMLR — Accepted by TMLR_

### Review · Reviewer_fVMN · 2024-03-04

**Summary Of Contributions:**

The paper conducts a reproducibility study of GNNInterpreter through investigating the four claims made in the original paper. The paper performs extensive experiments and suggest some claims are partially invalid.

**Audience:**

Yes

**Broader Impact Concerns:**

There's no broader impact concern.

**Claims And Evidence:**

No

**Requested Changes:**

1. As per weakness #1, the authors should include more detailed discussions and explanations on the methodology of GNNInterpreter.

2. As per weakness #2, the authors should make further attempts to analyze the reasons why GNNInterpreter does not show the expected benefits. One example is to investigate the cause of instability of GNNInterpreter through further experiments and ablations. It is expected that the authors could also provide some insights that would help develop a better model.

3. In Table 1, the performance of pretrained and retrained models on MUTAG dataset is significantly lower compared to the reported result in original paper. The authors should try to investigate the cause of such discrepancy.

4. It is suggested that the main findings and conclusions of this paper to be highlighted in Section 2, along with the claims of the original paper.

**Strengths And Weaknesses:**

Strength:
The paper conducts extensive experiments, both quantitatively and qualitatively verifying the efficacy of GNNInterpreter.

Weaknesses
1. Introduction to methodology of GNNInterpreter is not sufficient and it is difficult to understand the current discussion of how GNNInterpreter works. Also there many notations are presented without explanation. In particular, what is phi_c in equation (1) specifically? How continuous relaxation works? What is Omega, H, Theta in equation (2). How does GNNInterpreter solves the optimization problem in (2)? What is small omega_{i,j} in equation (5)?

2. The main conclusion from this paper is unclear apart from indicating the claims of the GNNInetepreter paper are partially invalid. Few insights are drawn and no guidelines on how to improve the model.

---

> ### Author Response · Authors · 2024-04-08
> **Addressing review**
>
> We have addressed each of the requested changes individually:
> - Requested Change 1: We extended the discussion about GNNInterpreter’s methodology in Section 4, better clarifying notations used in equations as well as adding additional information about the different components of the model.
> - Requested Change 2: We further investigated the cause of GNNInterpreter’s instability (see new section 6.4 - Analysis of training instability). Additionally, we drew new insights and guidelines on how to improve the model in the Discussion section.
> - Requested Change 3: We added a paragraph (in section 6.1) showing more insight into individual model performance on the MUTAG dataset. It shows that for some lucky seeds the performance does match that of the original paper.
> - Requested Change 4: We expanded Section 2 with our main contributions, both directly related to the original paper’s claims as well as our findings that go beyond the original scope.

---

### Review · Reviewer_2m7S · 2024-03-10

**Summary Of Contributions:**

This work attempts to reproduce GNNInterpreter paper and contextualise its claims, especially against the prior art in GNN interpretability (XGNN). The authors retrieve a codebase attributed to one of the GNNInterpreter's authors, and attempt a variety of experiments, bugfixes, and code variations to assess the contributions and claims in the work. A verdict is provided for the four claims originally stated in GNNInterpreter; not all of them have been easy to verify.

**Audience:**

Yes

**Claims And Evidence:**

No

**Requested Changes:**

See my general comments above. Typos, clarity fixes, additional experiments would all be welcome, but I also think the language in terms of claim rejection should likely be toned down or conceptualised better as well.

**Strengths And Weaknesses:**

First, I must remark that, while I am an expert in GNNs, I am not an expert in interpretability, or deeply familiar with the work being reproduced; I am hence taking all of the results from the presented reproducibility study at face value, and expert opinions should also be included when making the final decision on whether to accept.

I appreciated that multiple versions of the code, bugfixes, package stability, and an additional dataset (REDDIT-BINARY) are all taken into account -- though I would argue that what we learned from the new dataset is relatively minimal, given that the authors did not have the computational resources to train many model variants. The authors also make efforts to study training dynamics variations which are not even discussed in the original paper and report their findings. All relevant computational environment parameters have been reported. These findings are all of value.

Additionally, I found the literature survey to be well executed and contextualises the work well. Some interesting references may be missing; consider e.g. GCExplainer (Magister et al.).

There are certain small issues and clarity fixes that would be useful to make:

* The Gilbert reference should likely be Gilmer et al.?
* The symbols in Equation 2 being optimised over (Theta, Omega, Xi, Q, P, H) are not explained properly in the text.
* The max node count hyperparameter is not discussed before the description of which values it is set to, and from the text I would assume it was an important one. The authors should elaborate on what this hyperparameter does.
* _'If the extracted rules and explanation graph reflect the GNN faithfully, it is expected for the GNN to misclassify the 8 fake motifs as the ground truth motif.'_ --> I'm not sure if the word 'misclassify' here is always an appropriate one to use. If the explainer method fully achieves its aims and an explanation graph it generates is very accurate, then the fake motif might actually be a valid one, and hence not "mis"classified?
* Table 3 abruptly switches notation from 'us-original' to 'us-them' -- why?

As a major concern, I find it a bit dubious to base a full acceptance or rejection of a claim on a single dataset, especially a dataset like MUTAG. I am not too familiar with MUTAG's use for explainability, but in generic GRL applications, it is well known that MUTAG is oversaturated and utterly unreliable for making any kind of salient model comparisons.

Lastly, in summary, I find that the work makes a good effort to provide cautionary tales and practical advice to researchers and practitioners keen to extend GNNInterpeter, especially if they are doing so via the publicly available codebase. However I must say that based on the evidence available, I cannot conclude that the repository used corresponds to how GNNInterpreter's results were obtained. The codebase was not linked in the main GNNInterpreter paper, and no response from the authors of GNNInterpreter asserting the appropriateness of that codebase has been obtained. For all we know, the public codebase might be a highly-unstable fork of some private code.

I do agree with the authors that they have made a solid effort to reproduce with the information given, but I would be a lot more cautious in saying which claims can be reliably confirmed or rejected, or which implementations correspond to the actual paper. It might be useful to temper these claims and make them slightly toned down (e.g. _'we find evidence to reject Claim X within the experimental codebase found at..., over the ... datasets, with caveats ...'_ rather than _'we reject Claim X'_).

---

> ### Author Response · Authors · 2024-04-08
> **Addressing review**
>
> The following improvements/changes have been made according to the mentioned points:
> - We edited our conclusion and made our claims less strong, focusing more on the fact that our results support/contradict the claims instead of fully confirming or denying them.
> - We changed the order and wording surrounding claim 1, to emphasize that we tried out different variations of implementations and did not always directly follow the (unofficial) github implementation.
> - We added a paragraph in methodology explaining why the github implementation was chosen and how it still allows us to draw conclusions on GNNInterpreter. (Section 5)
> - We have addressed all small issues/clarity fixes:
>     - We included the missing reference to GCExplainer in the literature survey and we fixed the incorrect reference (Gilbert).
>     - We added a more detailed explanation of the symbols in Equation 2.
>     - We added some more clarification to the Verification Study in section 5.4. The fake motifs are manually created by qualitatively extracting rules from explanation graphs. These motifs include common features with the ground truth motif, but can never be the same motif as the ground truth since they are crafted manually.
>    - We fixed the inconsistent wording in Table 3.
>    - We have also conducted additional experiments, namely a comparative analysis on the MUTAG dataset with GNNExplainer and further investigations of the GNNInterpreter’s training instability. We have also provided a more detailed discussion on potential reasons and guidelines for improvement.

---

> > ### Comment · Reviewer_2m7S · 2024-04-10
> >
> > Thank you for covering these points. They address my concerns appropriately and I have no follow-ups.

---

### Review · Reviewer_vL3S · 2024-03-11

**Summary Of Contributions:**

The paper provides a detailed examination of the GNNInterpreter model through the replication of original experiments and the addition of new ones on an extra dataset. It focuses on four main claims about GNNInterpreter’s ability to generate faithful, realistic explanations across various data types, produce class-representative explanations, and reduce training time. The authors conduct both quantitative and qualitative analyses, testing the model on multiple datasets, including synthetic and real-world examples, to evaluate its performance and explainability.

**Audience:**

Yes

**Broader Impact Concerns:**

Np broader impact concerns.

**Claims And Evidence:**

Yes

**Requested Changes:**

See the weaknesses above.

**Strengths And Weaknesses:**

Strengths:

1. Comprehensive evaluation.

The paper thoroughly evaluates GNNInterpreter by replicating original experiments and extending them to include a new real-world dataset. These experiments provide comprehensive evaluation of the capabilities of the model.

2. Detailed analysis of claims.

Each claim made by the original paper is checked in detail. The authors not only test these claims but also provide an in-depth analysis of whether they hold under different conditions, adding credibility to their findings.

3. Quantitative and qualitative assessments.

The inclusion of both quantitative metrics and qualitative analysis (such as visual inspection of explanation graphs) offers a multi-faceted view of the GNNInterpreter's performance.

4. Transparency and reproducibility.

By using the open-source implementation and detailing their methodology, including data preprocessing and hyperparameter settings, the paper promotes transparency and reproducibility in research.

Weaknesses:

1. Lack of novelty in terms of methodology.

One weakness of the paper is the novelty. The study primarily focuses on reproducing existing experiments and validating the claims made by the original GNNInterpreter paper. However, the paper does not introduce new models, methodologies, or theoretical insights. While the reproduction of results is important fir ensuring that findings are reliable and reproducible, this approach lacks the introduction of innovative concepts or techniques that push the field forward.

2. Limited exploration of alternative approaches.

The paper does not compare GNNInterpreter with a broad range of existing explainability techniques beyond XGNN. Including more comparative analyses with other state-of-the-art explainability methods could have enriched the findings and offered a more comprehensive view of where GNNInterpreter stands in the landscape of GNN explainability solutions.

3. Generalization to other GNN models.

The reproducibility study is conducted with specific GNN models and architectures. The paper does not extensively explore the performance and applicability of GNNInterpreter across a diverse range of GNN architectures. Investigating how well GNNInterpreter generalizes to other models could highlight its versatility or reveal additional limitations.

---

> ### Author Response · Authors · 2024-04-08
> **Addressing review**
>
> Please refer to our comments below addressing each weakness:
>
> Weakness 1: It is essential to emphasize that the primary objective of our study was to rigorously validate and verify the findings of the original paper, not to introduce new ideas. Our focus lay on replicating the original experiments and confirming the robustness of the authors’ results. While introducing novel models and methodologies is valuable in advancing the field, it deviates from the core purpose of our reproducibility study. Furthermore, we found it imperative to be able to first reproduce all the claims made in the original paper, before attempting to extend on the original work. Since we were not able to accept most of the claims, our focus went to performing a thorough analysis on the model’s training instability. With additional experiments that went beyond the original study, we found that the GNNInterpreter works better with larger graphs and struggles with smaller graphs.
>
> Weakness 2: When it comes to exploration of alternative approaches, we have now conducted additional experiments using GNNExplainer. The implementation of this model was fairly old (3 years), therefore setting it up was not as easy as we envisioned. However, we managed to get interesting results on the MUTAG dataset. These can be viewed in section 6.3 - Results beyond original paper.
>
> Weakness 3: Regarding generalizing to different GNN models, we figured it was more important to first test the most basic GNN architecture. From our results we conclude that even for this simpler model there were still datasets where GNNInterpreter does not perform well. For this reason, we did not pursue any experiments with other models, as we reasoned that it would be more fruitful to first discover what exactly the limits are when using these simpler models and how it could be improved. Then afterwards, when sufficient improvements to the technique have been made, more advanced models can be tested.

---

> > ### Comment · Reviewer_vL3S · 2024-04-30
> > **Official Comment by Reviewer vL3S**
> >
> > Thanks for the response, which has addressed most of my concerns.

---

### Comment · Action_Editor_4THu · 2024-03-19
**Author response deadline postponed by two weeks**

Hello,

This message is to inform that per the authors' request, the deadline for submitting their response has been postponed by two weeks.

Best,

AE

---

### Decision · Action_Editor_4THu · 2024-05-02

**Recommendation:** Accept as is

**Comment:**

The submission contributes an effective reproducibility study of GNNInterpreter.  Reviewers had some reservations, mostly notably around the scope of the claims being made, but the authors addressed these concerns.  The main drawback of the work is that it does not introduce fundamentally new ideas or methodologies.  However, given the nature of TMLR, this drawback should not be an impediment for publication.  I believe reproducibility are important and am happy to recommend acceptance.

**Audience:**

The topic of the work (GNN explainability, and GNNInterpreter in particular) is highly relevant to the TMLR community.

**Claims And Evidence:**

The submission properly supports its claims.